# HGF/MET Signaling in Malignant Brain Tumors

**DOI:** 10.3390/ijms21207546

**Published:** 2020-10-13

**Authors:** Elizabeth Qian Xu Mulcahy, Rossymar Rivera Colόn, Roger Abounader

**Affiliations:** 1Department of Microbiology, Immunology & Cancer Biology, University of Virginia, Charlottesville, VA 22908, USA; qx3k@virginia.edu (E.Q.X.M.); rr5fr@virginia.edu (R.R.C.); 2Department of Neurology and the UVA Cancer Center, University of Virginia, Charlottesville, VA 22908, USA

**Keywords:** HGF, MET, malignant brain tumor, brain metastases

## Abstract

Hepatocyte growth factor (HGF) ligand and its receptor tyrosine kinase (RTK) mesenchymal-epithelial transition factor (MET) are important regulators of cellular processes such as proliferation, motility, angiogenesis, and tissue regeneration. In healthy adult somatic cells, this ligand and receptor pair is expressed at low levels and has little activity except when tissue injuries arise. In cancer cells, HGF/MET are often overexpressed, and this overexpression is found to correlate with tumorigenesis, metastasis, and poorer overall prognosis. This review focuses on the signaling of these molecules in the context of malignant brain tumors. RTK signaling pathways are among the most common and universally dysregulated pathways in gliomas. We focus on the role of HGF/MET in the following primary malignant brain tumors: astrocytomas, glioblastomas, oligodendrogliomas, ependymomas, and embryonal central nervous system tumors (including medulloblastomas and others). Brain metastasis, as well as current advances in targeted therapies, are also discussed.

## 1. HGF/MET Signaling

Hepatocyte growth factor (HGF), also known as scatter factor (SF), is a secreted protein that is involved in paracrine cellular signaling. It was discovered serendipitously, in 1984, in an effort to understand the extraordinary regenerative properties of the liver [1,2]. Prior to this discovery, SF, later resolved to be the same as HGF, was purified from fibroblasts and shown to promote epithelial cell motility and tissue morphogenesis [3,4]. A few years later, the proto-oncogene mesenchymal-epithelial transition factor (MET) was identified as the receptor for HGF [5]. 

Structurally, HGF is a 103 kDa soluble heterodimer, comprising an α- and a β-chain. MET is a 170 kDa glycosylated membrane protein, comprising an extracellular ligand-binding region, a single-pass transmembrane domain, and a catalytic intracellular domain [6]. 

When active HGF binds the extracellular portion of the β domain of MET, it causes receptor dimerization followed by the downstream auto-phosphorylation of two tyrosines located on the intracellular portion of the β domain (Y1349 and Y1356) [6]. This leads to the recruitment and binding of various important Src homology-2 (SH2) domain-containing proteins [7,8,9]. These proteins include phosphatidylinositol 3-kinase (PI3K), phospholipase C gamma (PLCγ), the non-receptor tyrosine kinase SRC and signal transducer and activation of transcription 3 (STAT3), adaptor proteins such as growth factor-bound protein (GRB2), GRB2-associated binding protein 1 (GAB1), and SH2 domain-containing transforming protein (SHC). From this, several important downstream pathways can be activated. The SOS-RAS-RAF-MEK-ERK-MAPK pathway is activated by GRB2. GAB1 is a scaffold protein that allows for the recruitment of additional signaling proteins such as PI3K, SHP2, and p120RasGap. Cell survival and proliferation are regulated by the RAS-MAPK and PI3K-AKT pathways as well as the transcription factors that are activated, STAT3 and NF-κB. Cell motility and migration are also activated by the GRB2-RAS-RAC1/CDC42-PAK and GAB1-CRK-C3G-RAP1 pathways. HGF can also form a ternary complex with MET that leads to cytoskeleton reorganization through the recruitment of proteins such as N-WASP, ARP2/3, cadherins, paxillin, focal adhesion kinase, and integrins. 

As a result of this wide network of signaling molecules being activated upon HGF/MET binding, this receptor–ligand interaction is heavily involved in several important biological processes including embryonic development, tissue regeneration, and wound healing in normal adult cells. 

### 1.1. Embryonic Development

Rodent models have been used to study HGF and MET functions in early embryonic development. For example, studies have found that mice lacking HGF and MET genes die in utero from severely impaired placentas and other organs [10]. In the mouse embryo, HGF is highly expressed in the limb bud, and mutant mice are not able to form the skeletal muscles of the limb and diaphragm [11]. HGF is also critical to the development of the nervous system, functioning to spatially direct axonal development for sensory, sympathetic, parasympathetic, and cortical neurons [12,13]. In addition to neuronal development, HGF is involved in normal glial development based on experiments done during postnatal development in the rat [14]. Organ culture experiments have shown that HGF and MET regulate mesenchymal and epithelial interactions and are expressed on the two cell types, respectively [15]. The paracrine interactions subsequently drive epithelial growth, morphogenesis, and differentiation through the differential spatial expression of HGF and MET [16]. 

### 1.2. Tissue Regeneration and Wound Healing

In addition to hepatic regeneration, which is the initial context in which HGF was first isolated and discovered, HGF and MET expression is also observed to be upregulated in injured organs such as the kidney, lung, heart, stomach, intestines, etc. [17]. This upregulation promotes tissue regeneration and wound healing in adult tissues [18]. Various animal studies have shown that endogenous HGF/MET is crucial to organ regeneration through treatment with blocking anti-HGF IgG antibodies and subsequently finding proliferation to be suppressed and organ repair or wound healing to have failed [17,19]. Because organ injury is caused by apoptotic events on the cellular level, one explanation for the role of HGF in tissue regeneration is that it can block apoptosis [20]. This occurs through the inhibition of caspase-3 or the induction of molecules such as Bcl-xL, which is known to be cytoprotective [21,22]. While HGF is a potent inhibitor of apoptosis in certain organs, it can also facilitate apoptosis in other organs. In this case, HGF signaling induces proteases that breakdown extracellular matrix scaffolding through activating matrix metalloproteinases (MMPs) such as membrane-type MMP and MMP-9 [20]. This activation is also seen in cancer cells and helps to promote spreading and invasion. 

As described in the previous section, HGF is involved in the development and survival of a variety of different neurons. As such, it has also been implicated in neuronal therapy in amyotrophic lateral sclerosis (ALS) [23]. Mice models have shown that neuronal overexpression of HGF leads to reduced motor neuron death and axonal degeneration [24]. 

Given the importance of this ligand/receptor pair, especially in the context of nervous system growth and development, it is no surprise that this and its many associated signaling pathways are hijacked by cancer cells for growth and survival. The following sections will first give an overview of malignant brain tumors (Section 2) and then discuss HGF/MET signaling in this specific dysregulated context (Section 3). 

## 2. Malignant Brain Tumors

In the United States, at any given time, more than 76,000 people are living with brain and other central nervous system (CNS) tumors [25]. In one year’s time, more than 19,000 people will have died from the malignant forms of these cancers, the most common being glioblastoma. Nearly 90,000 people will be newly diagnosed with a brain tumor each year. White men in their 80s are the most susceptible group within the world population. This has been hypothesized to be attributable to hormonal differences, genetic differences, and micro-environmental changes in the cell of origin [26]. Despite significant understanding and advances over the past half-century, malignant brain tumors remain largely incurable [27]. There are several reasons for this. 

First, malignant brain tumors are genetically heterogeneous and contain a complex array of somatic genetic alterations. This characteristic of these tumors can significantly weaken the effectiveness of monotherapies. Some examples of genetic alterations are recurrent mutations in the isocitrate dehydrogenase genes 1 and 2 (IDH1 and IDH2), receptor tyrosine kinase (RTK) amplification associated with genetic abnormalities (as this review will focus on in terms of HGF/MET), mutations in the promoter region of the telomerase reverse transcriptase (TERT) gene, mutations of the canonical “guardian of the genome” tumor protein p53 (TP53), etc. [28,29,30]. Oftentimes, more than one of these alterations are present in the tumor and give rise to a heterogeneous population of cells. 

Second, the brain is an immune privileged organ and is protected by the blood and brain barrier (BBB) which results from tight junctions formed between endothelial cells that make up the vasculature of the brain’s outer layer [31]. This can be a great obstacle for most available, Food and Drug Administration (FDA) approved drugs as only small molecules can cross [32,33]. More details on current therapies as well as chemical and physical methods to overcome BBB semi-permeability are presented in Section 6.3. 

Third, surgical resection within the brain is difficult given the confined space and its essential contents. Studies on the effect of surgical resection for low-grade gliomas showed that patients whose tumors were ≥90% resected by volume had a 91% chance of being alive eight years post diagnosis [34]. This is contrasted with patients whose tumors were <90% resected by volume and had a 60% chance of being alive eight years later. These studies have been adjusted for age, tumor location and subtype, and the respective Karnofsky Performance Scores (KPSs) of each patient. The KPS is a measurement of the patient’s ability to carry out ordinary tasks and is reflective of the impact of the disease [35]. In other words, although surgery can greatly improve disease prognosis, it does not entirely remove the tumor and does not cure the patient. 

Fourth, diagnosis is complicated by the often vague symptoms that present and the expense of imaging using magnetic resonance imaging (MRI). This is because symptoms are often convolved with other pathologies. For example, up to 80% of patients may present seizures, 30% present headaches, and 15% present morning nausea and vomiting [36]. Many of these symptoms are often attributed to other more likely diseases.

All of that said, there is a great need for treatment options for malignant brain tumors. Targeting RTKs is a useful approach because of how commonly the RTKs become dysregulated and thereby promote cancer growth and survival. The following sections will focus on the RTK MET and its ligand, HGF. As discussed in Section 1, this signaling pair is responsible for various aspects of normal cell growth, development, and tissue regeneration. Because of this, it is also important in cancer as these properties are used by cancer cells for growth, proliferation, invasion, and survival. The discussion of malignant brain tumors will include both primary tumors (gliomas, embryonal CNS tumors) and secondary tumors (brain metastasis). 

### 2.1. Primary Malignant Brain Tumors

Gliomas make up nearly 80% of all primary malignant brain tumors [37]. This review focuses on astrocytomas, glioblastomas, oligodendrogliomas, ependymomas, and embryonal central nervous system tumors (including medulloblastomas and others). See Figure 1 for a general summary of the different primary and secondary malignant brain tumors, their nomenclature-derived cell of origin, and real-life MRI images. 

Astrocytomas derive their name from astrocytes, a star-shaped cell that is the most abundant glial cell type in the CNS [38]. They help support neuronal cells by mediating synapses and providing nutrients. The cells of origin of astrocytomas are probably neural stem and/or oligodendrocyte progenitor cells in mouse models [39]. Glioblastomas arise from several different cell types. Over the past many years, the cells of origin for this particular tumor type has been the source of dissension and heated discussion for many. While its name is derived from glial cells which are the resident immune cells of the brain, glioblastomas likely arise from neural stem cells or perhaps even oligodendrocyte progenitor precursor cells [26,40]. In fact, many studies, prompted by this heated discussion, have given rise to a wealth of evidence showing the presence of glioblastoma stem cells in both mouse models as well as human tumors [41,42,43,44]. One of the most cited studies showing this was performed by Liu et al. in 2006 [43]. In this research paper, the authors used sorting methods to separate different populations of cells from primary cultured cell lines established from glioblastoma patient tissue samples. They found that more than 10% of the cells from the human tumors harbored the CD133 marker, a major stem cell marker. These cells were further isolated and shown significant resistance to chemotherapy. Furthermore, the expression of this marker was significantly higher in recurrent glioblastoma tissue samples of patients compared to the same patient’s newly diagnosed tumors. In 2011, research from John Laterra’s group showed that MET is activated and functional in glioblastoma neurospheres that harbor a large population of glioblastoma stem cells, and in fact, MET expression correlates with and induces the expression of stem cell markers and reprogramming transcription factors [44]. This further underscores the role of MET in brain tumor progression.

Oligodendrogliomas, despite the name, likely also arise from neural stem cells or glial progenitor cells [45]. Ependymomas develop from ependymal cells that make up the ventricular system of the brain [46].

Embryonal central nervous system tumors are formed from embryonic cells in the brain after birth [47]. Brain tumors in children are the second most common type of cancer in children after leukemia. There are two main types of central nervous system embryonal tumors: medulloblastomas and non-medulloblastoma embryonal tumors. Medulloblastomas arise in the cerebellum, likely from granule progenitor cells [48]. It is the most common type of primary brain tumor in children. Non-medulloblastomas are rare and include tumor types such as pineoblastoma, atypical teratoid/rhabdoid, and others [49,50].

### 2.2. Secondary Malignant Brain Tumors (Brain Metastasis)

Almost every malignant primary tumor is capable of metastasizing to the brain [53]. However, the incidence of brain metastases is difficult to quantify because some remain asymptomatic or are ignored because the patient is often severely ill with advanced primary disease. Autopsy studies suggest that as many as 30% of adult cancer patients with systemic malignancies have brain metastases at the time of death [54]. In adults, lung, breast, and melanomas account for 75% of the primary tumor of brain metastases [55]. In children and young adults, sarcomas such as Ewing’s sarcoma and germ cell tumors are the most likely to metastasize to the brain [56]. 

## 3. HGF/MET in Brain Tumors

Under physiological conditions, regulated HGF and MET signaling is crucial for embryonic development and tissue regeneration. When dysregulated, HGF/MET signaling promotes tumor progression and angiogenesis in many cancers including brain tumors (Figure 2). 

In the early 1990s, the first studies of HGF in cancer demonstrated the invasive properties of HGF through co-culture experiments of stromal fibroblasts and human oral squamous cell carcinoma [57]. The cancer cells become aggressive and invasive only when they are co-cultured with stromal fibroblasts. These latter cells secrete a protein that was later identified as HGF. Since then, the role of HGF/MET in cancer invasion has been demonstrated in a variety of cancer cells [58]. While HGF/MET in normal cells acts in a paracrine manner, HGF/MET signaling in cancer cells is often autocrine-mediated [59]. Mutations in the MET gene also induce tumorigenesis [60]. These mutations include missense mutations that cause either inherited or sporadic carcinomas. Amplification events are also abundant in MET-regulated tumorigenesis. 

In brain tumors, HGF and MET expression levels correlate with tumor grade in human gliomas [61,62,63]. When HGF is overexpressed in glioma cells, this functionally increases their tumorigenicity, growth, and angiogenesis [64,65]. Conversely, inhibition of HGF and MET expression leads to a decrease in in vivo tumor formation and growth of experimental gliomas [66,67]. Key experiments performed by Abounader et al. in the late 1990s demonstrated that when the in vivo expression of HGF is inhibited with synthetic RNA interference, this leads to substantial inhibition of tumor growth and prolonged survival in animals with human glioma xenografts [66,67]. 

At the cellular level, the HGF/MET pathway promotes tumorigenesis through stimulating cell cycle progressions, tumor cell migration, invasion, angiogenesis, and through inhibiting tumor apoptosis. For example, stimulation of human glioblastoma cells with HGF allows the cells to escape cell cycle arrest [68]. MET activation also leads to changes in key cell cycle proteins such as p27, phospho-Rb, E2F-1, and c-Myc. Additionally, while mechanisms that allow normal tissues to regenerate can be attributed to HGF/MET signaling, these same mechanisms are hijacked in tumors to protect the cells from DNA-damaging agents often used in chemo and radio-therapies by activating phosphoinositol 3-kinase-dependent and AKT-dependent anti-apoptotic pathways [69].

Since HGF/MET, in normal somatic cells and as discussed in Section 1, promotes cell motility and regulates a cohort of cytoskeletal proteins, this role is also hijacked in tumor cells through its interactions with the tumor microenvironment [70]. The activation of MET triggers Ras-dependent ERK1/ERK2 activation and STAT2 signaling. These signaling pathways in turn promote proliferation, survival, and migration of cancer cells. MET activation by HGF also leads to the activation of pathways that regulate epithelial–mesenchymal transition (EMT), a hallmark of cancer progression.

### Angiogenesis

Angiogenesis is the process by which new blood vessels are developed [59]. This process often accompanies tumorigenesis because rapidly growing cancer cells need nutrients and resources to build the growing colony. Angiogenesis occurs in several steps and involves at first, the breakdown of the pericellular and extracellular matrix to create a pathway for endothelial cells. Soon after, these endothelial cells form into a new blood vessel by forming the tubules and basement membrane of the vessel. The mechanisms involved here are the degradation of the ECM, vascular endothelial proliferation, migration, survival, and tubule formation. All of these mechanisms are regulated by HGF secreted by tumor cells and MET expressed on tumor endothelial cells. 

In the brain, for both developing and tumor cells, HGF/MET is highly expressed and functional. For example, neural endothelial cells that can form new blood vessels have been shown to express high levels of endogenous and active HGF [63]. In addition, immunofluorescence staining experiments have shown HGF/MET to localize more to regions of endothelial cells in the perivascular and vascular areas of higher grade brain tumors [61]. For brain tumors, in particular, HGF/MET induces endothelial cell proliferation and migration, expression of vascular endothelial growth factor (VEGF), and tubule formation [59].

Section 4 will focus on HGF/MET in specific types of primary and brain tumors and brain metastases. Primary malignant brain tumors that will be discussed are astrocytomas, glioblastomas, oligodendrogliomas, ependymomas, and embryonal central nervous system tumors. HGF/MET in brain metastases will be discussed subsequently. There are many similarities between HGF/MET signaling in the different malignant brain tumors.

## 4. HGF/MET in Primary Malignant Brain Tumors

Gliomas are the most common primary malignant brain tumors. They account for 80% of all malignant primary brain and central nervous system tumors, with an annual age-adjusted incidence of 5.55 per 100,000 in the United States [71]. Genetic alterations in gliomas occur frequently. Approximately 80% of low-grade gliomas harbor recurrent mutations in the isocitrate dehydrogenase genes 1 and 2 (commonly referred to as IDH1 and IDH2, respectively) [28,72]. Because these genetic alterations are so common, current classifications of gliomas, spearheaded by scientific discovery and adopted by the World Health Organization (WHO), have taken into account the IDH mutation status and have come up with three distinct categories of classification based on this genetic property: (1) IDH mutated with chromosome 1p/19q co-deleted; (2) IDH mutated without chromosome 1p/19q co-deleted; and (3) IDH wild-type [73]. Other gene amplifications or mutations have included RTK/RAS/PI3K, *TP53*, *ATRX* loss, and RB signaling pathways [74]. IDH mutations are seen in gliomas such as astrocytomas and oligodendrogliomas whereas IDH wild-type is present 90% of the time in glioblastomas [51]. The expression status of MET has been studied in the context of IDH mutations and has been found to be a useful prognosis marker, easier to evaluate than chromosome deletion status, correlating with prognosis prediction in IDH mutant astrocytomas and oligodendrogliomas as well as IDH wild-type glioblastomas [75]. HGF/MET are key determinants of malignancy in brain tumors, and their expression often correlates with the malignancy grade of gliomas [76,77]. The aberrant expression of MET in high-grade gliomas and embryonal brain tumors is associated with poor clinical outcomes. HGF binds to the receptor MET and induces several biological activities involved in cancer progression, such as growth, survival, motility, and metastasis [70,78]. According to the World Health Organization (WHO), there are four grades of gliomas and they are classified as low-grade gliomas (grades I and II) and high-grade gliomas (grades III and IV) [51].

### 4.1. Astrocytomas

Astrocytomas are classified by the WHO grading system by four histological grades of increasing malignancy [79]. Four malignancy grades are recognized with very different prognosis [80]. They include pilocytic astrocytoma (grade I), diffuse astrocytoma with IDH mutations (grade II), anaplastic astrocytoma with IDH mutations (grade III), and glioblastoma with either IDH mutations or wild-type (grade IV). Each has different degrees of aggressiveness to assess for similarities and differences at the level of individual genes, signaling pathways, molecular subtypes, and regulatory networks [79]. Malignant astrocytomas are associated with HGF overexpression [81]. HGF/MET signaling can stimulate various downstream signaling pathways in tumor cells, such as PI3K/AKT, JAK/STAT, Ras/MAPK, SRC, and Wnt/β-catenin [82], and it can enhance tumor malignancy by inducing biological processes such as tumor proliferation, invasion, and metastasis [76]. The binding of MET by HGF can induce structural changes in the protein, which can eventually lead to the activation of mitogen-activated protein kinases (MAPKs) [82]. Previous studies have also shown that an increase in activated AKT and MAPK, downstream of MEK, correlates with the progression of astrocytoma to glioblastoma [83].

### 4.2. Glioblastomas

Glioblastoma (grade IV astrocytoma) is the most common malignant primary brain tumor making up to 54% of all gliomas and 16% of all primary brain tumors, with an incidence rate of 3.19 per 100,000 persons in the United States and a median age of 64 years [84]. GBM is known by its highly mutated genome, which is associated with the dysregulation of many key signaling pathways involving growth, proliferation, survival, and apoptosis [85]. Aside from IDH mutations, which exist in about 10% of GBMs, there are three additional commonly deregulated pathways in glioblastoma: p53, retinoblastoma (RB), and RTK. In this context, the most commonly mutated one is the RTK/PI3K pathway, and there is approximately 88% of glioblastoma samples that harbor at least one genetic event in this core pathway [74]. MET is one of the deregulated RTKs, which promote malignant phenotypes in GBM. The MET pathway can increase levels of VEGFA and VEGFR2 on endothelial cells and promote proliferation, metastasis, and angiogenesis [86]. In addition, activation of the HGF/MET axis prevents apoptosis through activation of phosphatidylinositol-3-kinase (PI3 kinase) and subsequent AKT activation [87]. Furthermore, according to Cruickshanks et al., upregulation of the PI3K/AKT pathway in GBM leads to the growth and survival of uncontrolled tumor cells through the nuclear factor kappa-light-chain-enhancer of activated B cells (NFκB) that activates many cell survival and anti-apoptotic genes [88]. Another dysregulated pathway is RAS. By activating this pathway, MET can induce other signaling pathways such as MAPK, allowing tumor cells to grow and survive [88]. Moreover, upregulation of RTKs such as signal transducer and activator of transcription 3 (STAT3) can affect multiple signaling pathways in GBM [89]. It has been found that STAT3 and focal adhesion kinase (FAK) have a role in the promotion of GBM cell invasion and migration [90].

### 4.3. Oligodendrogliomas

Oligodendrogliomas are another type of glioma and are the second most common intraparenchymal brain tumor in adults [91], constituting 5–20% of all gliomas. According to the WHO classification, oligodendroglioma with IDH mutations and chromosome 1p/19q co-deletion is classified as a grade II. Anaplastic oligodendroglioma, differing from oligodendroglioma for histological reasons, with IDH mutations and chromosome 1p/19q co-deletion is classified as a grade III. They occur predominantly in adults, with a peak between 40 and 60 years of age and patients with low-grade tumors being slightly younger than those with high-grade, anaplastic tumors [92].

Ohba et al. showed that MET expression was correlated with progression-free survival in oligodendroglial tumors with IDH mutations [75]. However, at the cellular level, MET expression in oligodendroglial tumors seems to be lacking, and earlier reports state that MET positive cells were very rarely recognized in oligodendroglioma [75]. Furthermore, Pierscianek et al. investigated MET gain in diffuse astrocytomas and found that only 16% of oligodendroglioma had MET expressed [75,93].

### 4.4. Ependymomas

Another type of gliomas is ependymomas. Tumors of glial origin such as ependymomas have an incidence of 0.29 per 100,000, with approximately 240 cases in 2018 [94]. This type of glioma is classified in three different grades. Grade I ependymomas include sub-ependymomas, grade II are designated as ependymomas, and grade III tumors are called anaplastic ependymomas [95]. The five-year survival of pediatric ependymoma is approximately 57%, despite the advancement in therapies [96]. Aberrant expression of vascular endothelial growth factor receptor 2 (VEGFR-2), platelet-derived growth factor receptor β (PDGFRβ), the epidermal growth factor receptor family (ErbB1-4), and hepatocyte growth factor receptor (another name for MET) have been found in ependymomas [59,96]. Deregulated expression of RTKs and related growth factors such as VEGF, HGF, and PDGF can result in specific signaling that enhances tumor growth [97].

### 4.5. Embryonal Central Nervous System Tumors

Embryonal central nervous system (CNS) tumors include medulloblastoma (MB), atypical teratoid/rhabdoid tumor (AT/RT), pineoblastoma, and others [98]. Medulloblastomas are primary embryonal tumors of the central nervous system [99] and are classified by the WHO as a grade IV tumor [100]. Therapy for MB includes surgery, radiation, and chemotherapy, however, despite the advances in treatments, current five-year survival rates are approximately 60% [101]. Studies have shown that it is crucial to target important signaling pathways involved in medulloblastoma progression in order to have better therapeutic strategies [102]. HGF/MET is a key signaling pathway in these tumors as it has been implicated in the pathogenesis of medulloblastoma [101]. Li et al. showed that MET levels correlate with patient prognosis and that activation of the pathway has widespread and multi-functional tumor-promoting effects [102]. Moreover, HGF-activated MET paracrine signaling on endothelial cells can enhance their angiogenic activity [103]. Moreover, in the context of medulloblastoma, MET activation leads to the expression of proteins such as matrix metalloproteinases and vascular endothelial growth factor, which are known for their important roles in tumor promotion through angiogenesis [104].

## 5. HGF/MET in Brain Metastases

One of the most frequent malignant tumors of the central nervous system (CNS) are brain metastases (BMs), and about 20–40% of patients with cancer will develop BM in their clinical course [105]. Despite the use of various combinations of treatments such as surgery and radiotherapy, brain metastases are very difficult to treat. The response rates to single-agent chemotherapy are less than 10%, and treatment only slows but does not arrest or reverse disease progression [106]. It is becoming clear that the genetic background of a certain patient or a tumor should dictate its treatment regimen, and that targeted therapy against these tumor-specific alterations may be more efficacious (discussed in the next section) [106].

According to Demkova and Kucerova, one of the important signaling pathways that has been implicated in many cancers’ metastatic spread is signaling by HGF/MET [107]. HGF/MET has a key role in invasion, metastasis, and especially radio-resistance [108]. Moreover, the cellular functions regulated by this receptor and ligand pair include the entire process of metastasis such as migration, invasion, extravasation, anionic resistance in the vasculature, survival in unfamiliar microenvironments, and neovascularization [109]. HGF expression also plays a role in the different stages of metastasis. According to Mizuno and Nakamura, stroma-secreted HGF is required for cancer cells to infiltrate neighboring tissues, such as vascular beds, across the basement membrane [110].

Section 6 discusses current approaches to target HGF/MET for therapy, with a sub-section on methods to facilitate the crossing of the blood and brain barrier by HGF/MET-targeting agents.

## 6. HGF/MET Targeted Therapies

### 6.1. Monoclonal Antibodies

The significant role that HGF/MET plays in tumor progression and metastasis has made it a prime therapeutic target in oncology [111]. Monoclonal anti-HGF antibody functions by blocking HGF binding and thereby preventing MET activation. This can result in inhibiting the activation of signaling pathways downstream of MET [112].

For example, Onartuzumab/MetMAb is a monoclonal, monovalent anti-MET antibody designed by Genentech, Inc. It was created in an effort to overcome the obstacles of non-specific agnostic activity that can occur when divalent antibodies are used. It competes with HGF for binding to MET [74] and has been demonstrated to inhibit glioblastoma growth in preclinical testing [113]. MetMAb has been used in conjunction with bevacizumab but has not demonstrated effective therapy in a phase III clinical trial in lung cancer [114]. Another antibody, AMG102/Rilotumumab, is a human IgG2 monoclonal antibody designed by Amgen, Inc. [115]. This antibody has completed phase I and II clinical trials. This therapy, in combination with bevacizumab or temozolomide despite showing effectiveness in stabilizing disease progression, had failed in recent clinical trials due to increased risk of death [116]. Recently, ABT-700/Telisotuzumab, an anti-MET antibody developed by Laboratoires Pierre Fabre was tested in a phase I clinical trial with human patients with MET-amplified metastatic solid tumors [117]. The results showed that this antibody inhibitor was reasonably safe and had clinical activity. Similarly, LY2875358/Emibetuzumab is another anti-MET antibody developed by Eli Lilly and Company that has recently completed testing in a phase I clinical trial in human patients with non-small cell lung cancer [118]. As a result, some of the patients were able to achieve stable disease.

One major obstacle with monoclonal antibodies, although specific and effective, is the difficulty in drug delivery. Antibodies are big and often show decreased penetration across the BBB. Section 6.3 will discuss ways to overcome this obstacle with both physical methods as well as chemical modifications.

### 6.2. Small-Molecule Inhibitors

While antibodies are large and therefore cannot easily penetrate the BBB, small-molecules overcome the drug delivery obstacle by being small and often hydrophobic. As such, there has been a wide range of small-molecule inhibitors that have been tested in preclinical and clinical settings. They range from type I to VI based on a range of properties such as binding kinetics, structure, and degenerate binding of multiple RTKs [119].

Type I inhibitors are reversible ATP-competitive inhibitors with specific robust targeting of RTKs [120]. For example, AMG 337, developed by Amgen, is a type I small molecule inhibitor of the inactive activation loop of MET [121]. Once it binds, it locks MET into an inactive conformation and prevents downstream signaling. In mouse xenograft models, this inhibitor showed low toxicity and good toleration. Another common type I inhibitor is PF-02341066 developed by Pfizer, Inc. [87]. PF-02341066 was shown to be effective against MET-dependent growth, invasion, and survival and has been tested in phase II clinical trials for CNS and solid brain tumors. It is currently used as a commercially available treatment for non-small cell lung cancer. Recently, a phase I clinical trial was completed using PLB1001, a selective MET inhibitor, in patients with recurrent high-grade gliomas. This inhibitor showed a partial response in a small population of the patients with low significant side effects [122]. APL-101, another selective MET inhibitor, has shown anti-tumor effects in a variety of human tumors in mice models. Apollomics Inc. has announced that this inhibitor will be used in an upcoming clinical trial for glioblastoma patients with MET amplifications and fusions.

Type II inhibitors are also ATP-competitive inhibitors but they are less specific and can target multiple RTKs [120]. This is advantageous for the reason that certain cancers can have high levels of MET mutations and each MET mutant will need to be inhibited for the therapy to be most effective [123]. Cheng and Guo have reported INCB28060, produced by Selleck Chemicals LLC, type I/II, to be a potent and selective inhibitor of MET kinase and to show strong anti-tumor activity in MET-dependent mouse tumor models [74]. Furthermore, Liu et al. reported INCB28060 to be effective against not only MET signaling but other pathways such as EGFR and HER-3 that are regulated by MET [124]. This may help to reduce drug resistance in some patients [124].

Type III and VI inhibitors are non-ATP-competitive, meaning that they are allosteric or covalent inhibitors [119]. The first example of this to enter clinical trials is ARQ197 developed by Arqule, Inc. It acts to prevent HGF-dependent MET phosphorylation. It has been tested in phase I trials for patients with metastasis [87].

Additionally, there is a rare inhibitor of HGF called SRI 31215, developed by Eli Lilly and Company that has shown promise in preclinical studies and is now part of clinical trials to be tested in patients with colorectal cancer [125].

In addition to specific designs to target HGF and MET, there have been advances in the strategies to cross the BBB that are discussed in the next section. These strategies could be used in combination with the HGF/MET inhibitory drugs to enhance delivery and efficacy.

### 6.3. Delivery Strategies to Cross the BBB

The BBB makes drug delivery for many brain-related diseases difficult. Evolutionarily, it is designed to protect and provide a sanctuary for the brain [126]. As such, it acts as both a chemical as well as a physical barrier. Chemically, it comprises transport, metabolic, and enzymatic barriers. Physically, it is composed of an elaborate network of tight and adherens junctions between endothelial cells [127]. Because of these properties, strategies to cross the BBB have focused on chemical and physical stimuli aimed to disrupt the barrier and create temporary openings.

#### 6.3.1. Chemical Stimuli to Create Openings

Chemical methods to disrupt the BBB involve using toxins, vasoactive compounds, synthetic peptides, hyperosmolar solutions, detergents, etc. [126]. Many of these have only been investigated in an in vitro setting and are rather invasive since the degree of disruption is hard to control. For example, zonula occludens toxin, naturally excreted by bacteria, has been tested to induce a reversible, concentration-dependent opening of the tight junctions in cultured bovine brain capillary endothelial cells [128]. Histamine and VEGF have been investigated as vasoactive and inflammatory stimuli [129]. In brain tumors in rats, these compounds when used in combination with other therapies have demonstrated significant improvement in drug delivery and BBB permeability [130]. The synthetic peptide, Cereport is designed to mimic bradykinin, a vasoactive compound, and has demonstrated capabilities in modulating BBB permeability [131]. Hyperosmolar solutions have been used to encourage the reversible shrinkage of endothelial cells to enhance BBB permeability [126]. They are invasive because administration is done via arterial injection. Another compound that is commonly used is the detergent sodium dodecyl sulfate (SDS) [132]. SDS is a chemical surfactant and strongly interacts with lipids and proteins in the cell membrane, acting to solubilize these hydrophobic molecules.

#### 6.3.2. Physical Stimuli to Create Openings

Physical methods tend to be less invasive because they do not require surgical intervention for chemical injections but rather rely on energy-based fields such as acoustic, microwave, or electromagnetic [133,134]. The focused ultrasound technique uses acoustic energy that can be focused on a particular spot deep within the body while minimizing the effect on tissues elsewhere [135]. When this is coupled with synthetic microbubbles, the BBB can reversibly and temporarily open without acute neuronal damage or ischemia [136]. The bubbles serve two roles. First, they localize the effect of the ultrasound to the vasculature, and second, they reduce the energy required to open the BBB, and therefore, no surgery is needed and the ultrasound can be applied over the intact skull. Using magnetic resonance imaging (MRI) as a guide, this is a non-invasive approach to open specific regions of the BBB allowing for drug delivery [133]. As such, this technique has been used to deliver drugs to the brain of human cancer patients [137]. Electromagnetic field (EMF) pulses have also been used to increase the permeability of the BBB and have demonstrated efficacy in in vitro models [138]. The degree of permeability can be fine-tuned by changing the wave shape, frequency, and amplitude of the EMF.

Overall, focused ultrasound is one of the few commercially available strategies to induce changes to the BBB because of its low invasiveness [133]. When coupled with MRI, this strategy is both diagnostic as well as therapeutic. It can be used to enhance permeability to HGF/MET-targeting small molecules and large proteins alike as well as liposomes and nanoparticles. This physical stimulus to create openings in the BBB is a promising area of active research.

## 7. Conclusions and Outlook

In this review, we discussed HGF/MET signaling in both well-regulated as well as dysregulated contexts. Specific focus was placed on malignant brain tumors, current targeted therapies, and strategies to cross the blood and brain barrier. Because of the significance and importance of the HGF/MET signaling pathway, more research is needed here to clarify the connections to normal development as well as cancer growth and proliferation. This, in turn, will help facilitate more creative designs for antibodies as well as small molecule inhibitors, both of which are in great need to treat malignant brain tumors. HGF/MET therapies will likely be more effective in combination with cytotoxic therapies and other targeted molecular therapies.

## Figures and Tables

**Figure 1 ijms-21-07546-f001:**
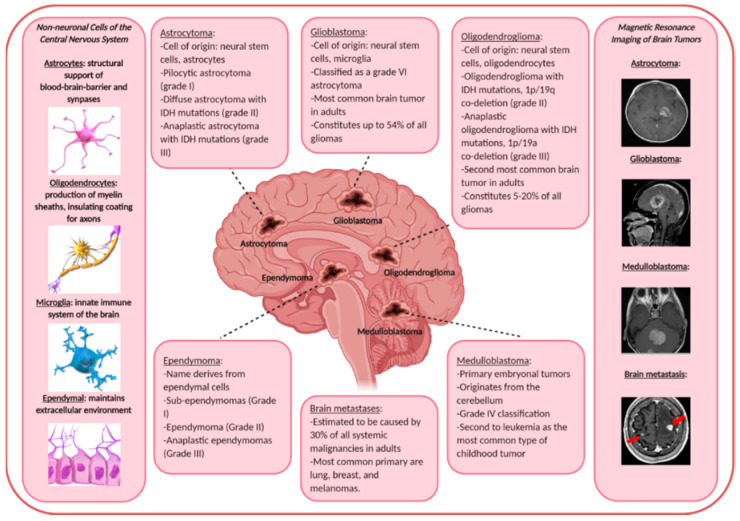
Summary of the different malignant brain tumors that are discussed in this review, their name-derived cell of origin, and real-life images taken from patients. Classifications are adapted from the World Health Organization [51]. Drawings of cells and patient tumor magnetic resonance imaging images are taken from the public domain [52].

**Figure 2 ijms-21-07546-f002:**
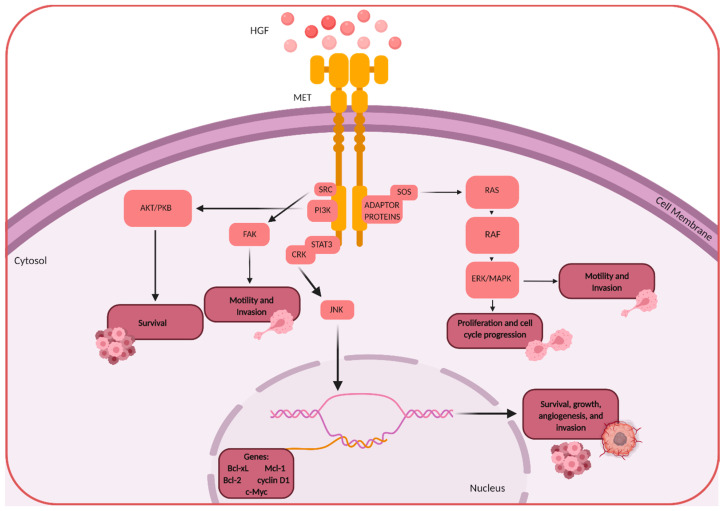
Select signaling pathways that are triggered by HGF/MET binding and activation, leading to important survival-promoting pathways in cancer.

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
