# Peer review of "HGF/MET Signaling in Malignant Brain Tumors"

_ijms, 2020, doi:10.3390/ijms21207546_

Round 1

Reviewer 1 Report

The Review “HGF/MET Signaling in Malignant Brain Tumors” by Elizabeth Qian Xu Mulcahy , Rossymar Rivera ColÏŒn , Roger Abounade, describe main molecular and functional studies carried on in the last two-three decades, devoted to investigate the role of the HGF/ MET complexes in controlling both physiological and pathological processes.

In particular, the Authors describe the studies that have shown how the HGF / MET system affects the progression of brain tumors. This article is well organized, well written and it can be very useful for clinicians preparing to handle brain tumors.

I have some concerns about the section n.6: Current Targeted Therapies. This section should be upgraded by including more recent studies in which new drugs targeting HGF/MET molecules have been tested in murine models and/or included in clinical trials.

Reviewer 2 Report

This Review discusses the role of the HGF/MET signaling in brain Tumors and translational potential of HGF/MET targeting. The Review provides a comprehensive coverage of the current state of knowledge about the molecular mechanisms of HGF/MET Regulation and involvement of this pathway in intrinsic brain Tumors and brain metastases. However, there some  structural insufficiencies that diminish the overall merit of this work with a suboptimal order and composition of individual sections requiring some improvements.

Specific comments:

Sections 2.1, 3 and 4 are all confined to the role of HGF/MET in primary malignant brain tumors and should be combined in one section

Section 2 “Malignant brain tumors”:

The concluding paragraph in Section 2 (lanes 128-133) is a bit out of this section’s context focused on the description of brain tumors types

Fig. 1 It would be more appropriate to refer to the WHO classification of Brain tumors

Lane 143: There is a wealth of evidence for glioblastoma stem cells in human tumors not only in mouse models.

Lane 145: The statement that term glioblastoma “is derived from microglial cells” is not correct and creates an erroneous impression that glioblastomas originate from microglia. Term “glioblastoma” reflects the fact that this is a tumor of the brain glia made of microglia and macroglia (astrocytes, Müller cells). In fact, glioblastomas categorized by WHO as grade IV astrocytomas make up more than half of all astrocytic tumors.

Section 3 “HGF/MET in brain tumors”:

lanes 174-182: description of HGF/MET in the context of non-brain tumors should be moved to section 1

lanes 184: “In brain tumors, HGF and MET expression levels are correlated with tumor grade in human gliomas[56-58].” i) correlate” instead of “are correlated”; ii) the following sentence “ The higher the levels, the higher the tumor grade.” Is redundant and should be removed

lane 186: “inhibition of HGF and MET expression leads to a decrease in in vivo tumor formation and growth of experimental gliomas

lane 211: confusing sentence “Similarly, in brain tumors, HGF/MET are expressed and functional in the developing brain and tumor vasculature.”

The discussion of HGF/MET role in brain tumors is restricted to angiogenesis, which is an important but not the only hallmark of glioblastomas. The role of HGF/MET in tumor cell migration, invasion, proliferation, apoptosis and interaction with the tumor microenvironment should be discussed as well.

Section 5 titled “HGF/MET in brain metastases” is very superficial and provides no detailed insights into the sub-topic it aims to discuss

Round 2

Reviewer 1 Report

The Authors have replied to all recommendations and the required information has been included. This revised version has been improved and deserve publication.